# The Effectiveness of Cyrene as a Solvent in Exfoliating 2D TMDs Nanosheets

**DOI:** 10.3390/ijms241310450

**Published:** 2023-06-21

**Authors:** Jaber Adam, Manjot Singh, Avazbek Abduvakhidov, Maria Rosaria Del Sorbo, Chiara Feoli, Fida Hussain, Jasneet Kaur, Antonia Mirabella, Manuela Rossi, Antonio Sasso, Mohammadhassan Valadan, Michela Varra, Giulia Rusciano, Carlo Altucci

**Affiliations:** 1Department of Physics “Ettore Pancini”, University of Naples “Federico II”, 80131 Naples, Italy; jaber.adam@unina.it (J.A.); fida.hussain@unina.it (F.H.); jasneet.kaur2@unina.it (J.K.); antonio.sasso@unina.it (A.S.); 2Department of Advanced Biomedical Sciences, University of Naples “Federico II”, 80131 Naples, Italy; manjot.singh@unina.it (M.S.); chiara.feoli@unina.it (C.F.); anto.mirabella@unina.it (A.M.); mohammadhassan.valadan@unina.it (M.V.); 3Italy National Institute of Nuclear Physics, Naples Section, 80126 Naples, Italy; 4Department of Pharmacy, University of Naples “Federico II”, 80131 Naples, Italy; avazbek.abduvakhidov@unina.it (A.A.); varra@unina.it (M.V.); 5Department of Precision Medicine, Università degli Studi della Campania “L. Vanvitelli”, 80138 Naples, Italy; mariarosaria.delsorbo@unicampania.it; 6Department of Agricultural Sciences, University of Naples “Federico II”, 80131 Naples, Italy; 7Department of Earth Science, Environment and Resources, University of Naples “Federico II”, 80131 Naples, Italy; manuela.rossi@unina.it; 8Istituto di Cristallografia—CNR, Via G. Amendola 122/O, 70126 Bari, Italy; 9ISASI-CNR, Institute of Applied Sciences and Intelligent Systems “Eduardo Caianiello”, 80078 Naples, Italy

**Keywords:** 2D-MoS_2_, 2D-WS_2_, LPE, cyrene, viscosity, DLVO

## Abstract

The pursuit of environmentally friendly solvents has become an essential research topic in sustainable chemistry and nanomaterial science. With the need to substitute toxic solvents in nanofabrication processes becoming more pressing, the search for alternative solvents has taken on a crucial role in this field. Additionally, the use of toxic, non-economical organic solvents, such as N-methyl-2 pyrrolidone and dimethylformamide, is not suitable for all biomedical applications, even though these solvents are often considered as the best exfoliating agents for nanomaterial fabrication. In this context, the success of producing two-dimensional transition metal dichalcogenides (2D TMDs), such as MoS_2_ and WS_2_, with excellent captivating properties is due to the ease of synthesis based on environment-friendly, benign methods with fewer toxic chemicals involved. Herein, we report for the first time on the use of cyrene as an exfoliating agent to fabricate monolayer and few-layered 2D TMDs with a versatile, less time-consuming liquid-phase exfoliation technique. This bio-derived, aprotic, green and eco-friendly solvent produced a stable, surfactant-free, concentrated 2D TMD dispersion with very interesting features, as characterized by UV–visible and Raman spectroscopies. The surface charge and morphology of the fabricated nanoflakes were analyzed using ς-potential and scanning electron microscopy. The study demonstrates that cyrene is a promising green solvent for the exfoliation of 2D TMD nanosheets with potential advantages over traditional organic solvents. The ability to produce smaller-sized—especially in the case of WS_2_ as compared to MoS_2_—and mono/few-layered nanostructures with higher negative surface charge values makes cyrene a promising candidate for various biomedical and electronic applications. Overall, the study contributes to the development of sustainable and environmentally friendly methods for the production of 2D nanomaterials for various applications.

## 1. Introduction

From the past decade, it is clear that nanoscience has played a significant role in revolutionizing the discovery of novel materials with unique properties and vast application areas. Since the discovery of graphene [1,2], two-dimensional nanomaterials (2DMs), such as graphene [3], transition metal dichalcogenides (TMDs) [4] and other nanomaterials (NMs) [5], have attracted enormous enthusiasm due to their astonishing physical and chemical properties. Due to their unique characteristics, 2DMs are suitable for a wide range of applications in electronics/optoelectronics [6], energy storage [7] and biological systems [8,9]. Among 2DMs, TMDs, such as MoS_2_ [10] and WS_2_ [11], have become promising substitutes for graphene due to their high conductivity, exceptional catalytic abilities, high charge-density-wave transition and good biocompatibility [12,13]. It has been found that 2D TMDs exhibit a band gap of around 1–3 eV that becomes layer-dependent: for example, MoS_2_ exhibits an indirect band gap in the form of a bulk-layered structure that transforms into a direct band gap of 1.8–1.9 eV [14] when it is machined in the form of nanosheets (NSs), becoming a semi-conducting layered material held together by weak van der Waals forces. Hence, the electronic structure of these materials can be tailored by machining their thickness in terms of the layer number of the fabricated NSs. This feature provides a unique and flexible platform to explore the potential of these NMs in view of diverse applications, such as, for WS_2_ NSs, in environmental remediation, energy conversion and nano-theranostics [15].

Exfoliation of various 2D TMDs with high surface-to-volume ratios has remained a challenge for material scientists until now because many of the physicochemical processes involved in their fabrication in a given solvent context are still unknown. Generally, the fabrication of NMs is categorized into two approaches: bottom-up and top-down. The former utilizes the chemical reactions between atoms/ions or molecules and is based on various techniques, such as hydrothermal and chemical vapor deposition [16], whereas the latter is based on mechanical methods that break the bulk into nanoparticles by overcoming the van der Waals interlayer forces. In the top-down approach, the NMs’ exfoliation is based on mechanical [17], solvent-assisted [18], chemically assisted [19] and electrochemical approaches [20]. 

Among these techniques, mechanical exfoliation results in high-quality monolayers, but it cannot be used at a large scale and does not allow systematic control over flake thickness and lateral dimensions. In the solvent-assisted approach, liquid phase exfoliation (LPE) is the most promising fabrication technique to date, as it allows production of large quantities of exfoliated NSs with good control over flake size and thickness, along with other advantages, such as the ease of producing composites and hybrid films (by just mixing two dispersions) [21,22]. The solvent plays a crucial role in this process, as it can affect the properties of the resulting materials, including their crystallinity, thickness and structural defects [23]. One of the ways to choose the best possible solvent is by matching its surface tension to that of the material [24]. This makes it possible to obtain higher stability for the fabricated dispersion.

Traditional solvents, such as N-methyl-2 pyrrolidone (NMP) and dimethylformamide (DMF), have been considered as the best solvents to exfoliate high-quality 2D NSs, but, on the other hand, these so-called “best” organic solvents are highly toxic; not environment friendly; very expensive compared to other commonly used organic solvents, such as NMP and DMF; and extremely difficult to post-process for applications. Additionally, these solvents are not at all suitable for sustainable chemistry processes, which has boosted the quest for potentially green novel NMs [25]. 

In this context, new aprotic, organic, green and eco-friendly solvents have been introduced into the scientific community for many applications [26]. Among these solvents, cyrene (known as 1S,5R)-6,8-dioxabicyclo [3.2.1]octan-4-one dihydrolevoglucosenone) is a pale yellow, dipolar, aprotic, green solvent with a mild ketonic odor derived in two simple steps from cellulose-containing biomass (cellulose to levoglucosenone and then to cyrene) [27]. It has been considered as the best-performing solvent in terms of toxicity [28], environmental persistence [29] and biocatalysis [26]. With physical properties such as high viscosity (14.5 cP at 25 °C), higher than that of NMP (1.67 cP at 25 °C) and DMF (0.92 cP at 25 °C), it plays an important role in exfoliating high-quality 2D TMD nanoflakes with fewer defects. Thus, inspired by these promising properties, several studies have shown the successful exfoliation of graphene in cyrene with high concentrations and better flake quality in terms of fewer defects [30], reporting the use of cyrene in the implementation of biocompatible inkjet printing and IoT applications [31]. 

As shown in one of our studies, the interaction of MoS_2_ with cyrene can be modeled using the DLVO, a simple approach that easily allows the optimization of the fabrication parameters of the NMs [32]. 

In the current article, we highlight—to the best of our knowledge, for the first time—the fabrication of MoS_2_ and WS_2_ NSs directly in cyrene using a modified LPE technique. This was achieved by pretreating the solution, using bath sonication and then, finally, exfoliating with probe sonication for a longer time. We analyzed the behavior and the quality of the 2D NSs produced in this extremely viscous solvent. The synthesized 2D TMDs were characterized using various techniques, including ς-potential, UV–visible and Raman spectroscopies and atomic force (AFM) and scanning electron (SEM) microscopies. The results of these analyses were used to evaluate the effectiveness of the LPE method using cyrene as a solvent in the preparation of high-quality MoS_2_ and WS_2_ 2D flakes. The findings of this study demonstrate the potential of using cyrene as an effective solvent for the synthesis of high-quality MoS_2_ and WS_2_ 2D NMs with minimal environmental impact and its potential usefulness as a green solvent in biomedical applications. 

## 2. Results and Discussion

### 2.1. Exfoliation of MoS_2_ and WS_2_ NSs

One of the bottlenecks that needs to be overcome at this stage of nanoscience is achieving sustainable production of high-quality 2D materials, such as MoS_2_, WS_2_, graphene and its analogues, to make them available at a large scale. To achieve this aim, LPE is considered as one of the best routes, being easily accessible and capable of harnessing the potential of solvents used to disperse various classes of 2DMs [33]. Various factors play an important role in deciding the quality of exfoliated products, such as the type/quality of the pristine material, the exfoliation efficiency, the dispersibility of the solvent and post-processing fabrication steps [23,34]. To date, NMP is recognized as the most efficient organic solvent for the dispersion of various 2DMs, but it is neither sustainable nor suitable for up-scaling production due to its adverse impact on the environment [35]. 

In the current study, we explored the potential of cyrene for the exfoliation of 2D-MoS_2_ and 2D-WS_2_ nanosheets under optimized sonication conditions [36]. So far, cyrene has only been used to exfoliate graphene for bio-inkjet and IoT applications, for which optimized fabrication conditions were utilized to obtain high-quality 2D graphene nanosheet dispersion [31,37]. 

Here, inspired by our previous study, we employed a modified LPE technique to fabricate MoS_2_ and WS_2_ nanosheets in cyrene [38]. The present case setup operated under controlled temperature during sonication, which made it possible to obtain very stable and concentrated samples of 2D TMD dispersions.

### 2.2. Optical Characterization: UV–Vis Spectra

LPE will always produce polydisperse nanosheets including both small- and large-sized 2D nanosheets. This also means that the final dispersion will contain 2D NSs of varied thicknesses, which can be separated in accordance with their size and thickness upon liquid cascade centrifugation. In this context, the acquisition of UV–Vis spectra is a quick approach to evaluate the average number of layers <N>, average size <L> and average concentration <C> for the exfoliated dispersion [38,39,40,41]. Indeed, within one sample/dispersion, UV spectra show characteristic, size-dependent spectral changes originating mostly from the absorbance and scattering components of the extinction, as the optical extinction spectra of small and large nanosheets include contributions from both the edge and basal plane effects, respectively [42,43]. 

Pure cyrene is not UV-transparent (Appendix A); therefore, the UV–Vis analyses of the exfoliated MoS_2_ NSs were performed by diluting the initial samples with water or methanol to reduce the strong solvent background in the 200–400 nm range. All the extinction bands in the UV–Vis profiles of the MoS_2_/cyrene: water mixture (1:1 *v*:*v*) had very low intensity and were hardly detectable at their maximum values (Appendix A). After decreasing the ratio of MoS_2_/cyrene:water down to 1:20, the resulting UV spectra showed significant decreases in all the extinction bands, which became no longer identifiable (Conversely, the UV–Vis profiles of the MoS_2_/cyrene: methanol 1:1 (*v*:*v*) mixture showed well-defined exciton bands (exciton A at 669 nm and exciton B at 610 nm; Figure 1A,B), which made it possible to calculate <N> for the production (N = 5.6; see Appendix A for more details) [44]. 

Furthermore, at around 345 nm, the minimum for the UV profiles of MoS_2_ NS dispersions generally appears [45], and the extinction spectrum at this wavelength, dominated by NS absorption phenomena, was used to calculate the NS average lateral size <L>. Nevertheless, the UV–Vis profiles of the MoS_2_/cyrene: methanol 1:1 mixture were still partly affected by the cyrene absorption background, with this minimum shifting from 345 to 334 nm (Figure 1C). The <L> value calculated for MoS_2_ from the absorption at 334 nm by using Equation (1) below [46] resulted in (121 ± 3) nm:(1)$$L\left(\mu m\right)=\frac{3.5{\mathrm{Ext}}_{B}/{\mathrm{Ext}}_{334}-0.14}{11.5-{\mathrm{Ext}}_{B}/{\mathrm{Ext}}_{334}}$$
where L is the average lateral size of the 2D NSs, ${\mathrm{Ext}}_{B}$ is the extinction value of exciton B in the UV–Vis spectra of the 2D TMD NSs and ${\mathrm{Ext}}_{334}$ is the extinction value at λ_min_ = 334 nm (in our results). This equation works well for nanosheets in the 70 nm < L < 350 nm size range, which is the typical size range for nanosheets produced by liquid-phase exfoliation. 

However, as the transmittance of UV radiation at this wavelength was still significantly low due to the cyrene absorption, we also measured UV spectra using 5:1, 9:1 and 17:1 methanol: MoS_2_/cyrene mixtures (Figure 1D). Herein, the transmittance became acceptable, but the convolution of the UV bands and the reduction in their intensity due to dilution caused the minimum in the 300–400 nm region to be no longer detectable. Therefore, we used the deconvolution function [47] (the Gaussian full width at half-maximum (FWHM) function) in Spectra Analysis (Jasco software, Spectra manager, version 2.15.01), which enabled detection of the minimum (Appendix A). In the case of the 5:1 methanol: MoS_2_/cyrene mixture, this minimum was detected at 336.0 nm, whereas, when using 9:1 or 17:1 methanol: MoS_2_/cyrene dilution levels, a very slight blue shift occurred (336.6 nm), thus suggesting that the cyrene background did not interfere with the UV absorption of MoS_2_ at these two dilution levels. The <L> values obtained using the parameters from the latter two deconvoluted UV spectra were in the 165–179 nm range. No UV bands were observed in the UV profiles of the WS_2_–cyrene dispersion diluted in water or methanol, which also used a final ratio between WS_2_–cyrene and water or methanol of 2:1. The results obtained from previous studies suggested that the exfoliation procedure adopted gave rise to the formation of WS_2_ quantum dots, for which confinement effects and a band gap transformation from indirect to direct occurred [48]. However, the use of cyrene, also in methanol-diluted mixtures, prevented the detection of the exciton edge band at 243 nm. Notably, differently from MoS_2_–cyrene, the WS_2_–cyrene mixture appeared transparent, similar to a solution rather than a suspension.

### 2.3. Microscopic Characterization: AFM 

WS_2_- and MoS_2_-based nanoflakes produced by exfoliation in cyrene were also characterized by AFM according to the protocol described in the Section 3. Figure 2A,C show representative AFM scans for WS_2_ and MoS_2_ nanoflakes, respectively, together with selected flake height profiles. Both images clearly reveal the presence of few-layer flakes. From the analysis of numerous AFM scans, the thickness distributions of both the WS_2_ and MoS_2_ nanosheets were obtained, as shown in Figure 2B,D, respectively. Notably, in both cases, a lognormal distribution was observed (highlighted by the fit curves of the histogram data), as expected for exfoliated nanoflakes [49].

In the case of WS_2_, the lognormal distribution peaked around 3 nm, and more than 50% of the observed nanoflakes had thicknesses < 4 nm, confirming the effectiveness of our approach in producing few-layer flakes. 

For the MoS_2_ flakes, the fitted curve peaked around 4 nm but was much wider, with most flakes (~35%) exhibiting thicknesses in the 4–8 nm range. Such a feature could have been due to somewhat incomplete exfoliation of the flakes with the employed protocol parameters. However, ~25% of the obtained flakes exhibited thicknesses < 4 nm, and ~60% of them were thinner than 8 nm; therefore, they are reasonably suitable for numerous bio-related applications. 

### 2.4. Spectroscopic Characterization: Raman Spectra 

Finally, the exfoliated materials were investigated using Raman analysis, a quite common and effective tool for the characterization of 2D materials, due to the quite rich chemical and structural information provided by their Raman spectra. In particular, in layered TMDs, the E^1^_2g_ and A_1g_ modes undergo a blue and a red shift, respectively, when passing from bulk crystals to monolayers flakes. This feature has been widely used in previous investigations to identify the number of layers in single exfoliated flakes [50,51,52]. In the present investigation, as the average nanoflake size was smaller than the spatial resolution of our system (in-plane and axial resolutions (PSF HWSHM) of ~0.3 and ~0.1 µm, respectively), it was not possible to assign the acquired spectra to single nanoflakes, nanoflake groups or clusters, and the obtained information had to be averaged in accordance with the confocal detection volume. 

In Figure 3, we show the mean Raman spectra for WS_2_ (A) and MoS_2_ (B) nanoflakes obtained by averaging 10 spectra acquired from different points of the samples. In the same figure, we also show the fitting curves for the Raman spectra using multipeak Gaussian functions. In particular, in the case of MoS_2_, the fitting was performed using two Gaussian peaks featuring the characteristic E^1^_2g_ and A_1g_ bands. As is well-known, the frequency shift Δν_MoS2_ = ν _A1g_ − ν _E_^1^_2g_ between these modes can be used to identify the number of layers in the nanoflakes. In our case, the Gaussian fitting allowed an accurate estimation of the peaks’ centers (~0.1 cm^−1^ error), which finally provided a Δν_MoS2_ of ~24.6 cm^−1^, corresponding to nano-structuring spanning three to four layers. This outcome was consistent with the range of nano-structuring indicated by UV–Vis extinction spectroscopy. 

Similar considerations could not be easily extended to the case of the WS_2_ flakes. As is clearly visible in Figure 3A, the WS_2_ Raman spectrum exhibited a complex pattern of overlapping peaks. This was partially due to a resonance effect for Raman excitation at 532 nm [53], which leads to complete overlapping of the 2LA(M) and E$\frac{1}{2g}$ modes. For this case, following the indications reported in [54], the WS_2_ spectrum was fitted using five Gaussian peaks. As can be seen, the fit quality was relatively high. Nevertheless, a wide error (~3 cm^−1^) was obtained for the position of the overlapping peaks in the 325 cm^−1^–360 cm^−1^ spectral region. This intrinsically hindered the determination of the number of layers in the WS_2_ flakes on the basis of the shift between the E^1^_2g_ and A_1g_ modes. Nevertheless, some conclusions could be drawn on the basis of the spectral position of the A_1g_ band only. In particular, previous investigations with few-layer WS_2_ flakes [53,54] found resonant Raman spectra exhibiting an A_1g_ band at 416.7 cm^−1^, which was consistent with the frequency of A_1g_ (~417.0 cm^−1^) for the spectrum in Figure 3A. Notably, this was also in agreement with the AFM-based outcomes, which revealed the few-layer character of the produced WS_2_ flakes.

### 2.5. Morphological Characterization: SEM 

We further proceeded to characterize the main morphological features of the produced 2D NSs in cyrene using SEM. We highlight that the cyrene matrix context of our nanosheets, which were indeed embedded into a cyrene film, formed several different structures in a single image. These structures could be viewed at a large spatial scale (100 µm steps in Figure 4A). Once diverse smaller domains with different features were identified within a single large domain, we zoomed into the identified smaller domains of interest to characterize different subsets of deposited NSs in each smaller domain (e.g., 100 nm and 200 nm steps in Figure 4B–F).

The SEM analyses were carried out with the MoS_2_ and WS_2_ NS flakes absorbed in the cyrene solid solution drop-casted on regular (Figure 4A–F) and irregular (Figure 5A–D) glass supports. These investigations allowed us not only to characterize the dimensions of the MoS_2_ and WS_2_ flakes but also to see how they were dispersed inside the cyrene solution. Therefore, we could observe the different textural and morphological features of the cyrene film with MoS_2_ and WS_2_ NSs, as below described. Visualizing the NSs in their solvent matrix context was important in view of the possible bioapplications of our liquid dispersion, mostly in the case of the cyrene solvent, which, though organic, is nevertheless biocompatible [36] and thus promising in this direction.

In Figure 4A–F, we report the characterization of 2D-MoS_2_ and 2D-WS_2_ NSs exfoliated inside cyrene and deposited onto a very regular silicon substrate. In Figure 4A, which shows the overall film texture and morphology, the presence of 2D-MoS_2_ NSs can be seen in the cyrene at the rim and at the core of the drop-casted and subsequently dispersed solid solution on the regular glass support. At the rim, the cyrene film was thicker, folded and wrinkled, showing an irregular texture. In proximity to the rim, there were many coffee rings of different sizes. The irregular texture at the rim was linked to the presence of MoS_2_ NSs totally absorbed in the cyrene film with very different lateral sizes, the mean being <L> = 122 ± 44 nm from 30 single measurements (Figure 4C) scattered in a chaotic way. 

At the core of the drop-casted and subsequently dispersed solid solution and at the center of the coffee ring structure [55], the film layer thickness was lower and the texture was more homogeneous (Figure 4E). The cyrene film rarely showed folds and wrinkles and the presence of coffee rings was reduced, but there were fractures. Here, the flakes had very small <L> values, the mean being <L> = 26 ± 7 nm over 30 single measurements. However, nearby, we found domains where <L>, calculated from the flakes present in these domains, was even lower than 20 nm and at the threshold for nanoflakes to be considered as quantum dots. The flakes at the core of the cyrene film appeared as isolated and/or grouped, but they were more evenly distributed than at the rim. The intensity of the 2D-MoS_2_ NS signal was quite low in comparison to that obtained from deposition in other dipolar aprotic solvents (details in Table 1), as the high viscosity of cyrene hindered the visualization of morphology, even after diluting the dispersion. 

It is important to note here that, in Table 1, we report the 19–38 nm values as the minimum and maximum 2D-MoS_2_ NS lateral sizes, respectively. This range of sizes was only ascribed to the cluster of 2D-MoS_2_ NSs dispersed and deposited onto the core of the glass substrate for SEM measurements and not to the entire production of NSs. To clearly visualize the morphology of the 2D nanoflakes dispersed in cyrene, we measured deep into the thin film deposited onto the substrate by analyzing the nanomaterial in the specific regions of the core and rim (Figure 4). In contrast, <L> = 121 nm represents the overall average size for the production as estimated from the UV–Vis absorbance of the whole dispersion.

The NSs appeared to be non-homogeneously distributed only in the SEM images shown in Figure 4C. In total, the morphology of the thin film was homogeneous at the core and irregular at the rim; additionally, small conchoidal fractures appeared in the same image. The flakes not only showed different sizes but were differently dispersed at the rim and the core of the drop-casted solid solution.

As shown in Figure 4B, the 2D-WS_2_ NSs in cyrene film turned out to be more homogenous than in the case of the 2D-MoS_2_ NSs. Here, we also observed the formation of a thin film for the solvent, which was less folded and wrinkled near the rim. The coffee rings were larger than those for the 2D-MoS_2_ NSs. The surface of the thin film looked homogeneous right at the center of the regions forming coffee rings, the only irregularities being the folds near the rim, the roughness profile and the punctual defects observed far from the rim film. The texture showed strong regularity in the dispersion of flakes both at the rim and the core in comparison to 2D-MoS_2_ NSs. The irregular texture was due to the presences of punctual defects, where the flakes showed different sizes and were stacked on the defect surface. In this area, shown in Figure 4D, it was possible to observe the presences of WS_2_ NS aggregates with average lateral sizes <L> = 43 ± 16 nm. The size ranged between 29 nm and a maximum of 69 nm. The flakes were only partially absorbed in the cyrene layer coming up over the film surface. 

Finally, at the core of the solid solution and in the center of the coffee ring, the film had a regular texture and the WS_2_ NS flakes were homogeneously distributed and absorbed in the thin cyrene layer (Figure 4E). The sizes of the nanoflakes here were between 11 and 21 nm, and they rarely formed aggregates. Statistical sampling of 30 detected single nanoflakes resulted in an average <L> = 13 ± 5 nm, which definitely indicated the presence of many 2D-WS_2_ quantum dots in the preparation. The morphology of the thin film was more homogeneous both at the core and the rim compared to what was observed for the MoS_2_ NS cyrene solution; additionally, small conchoidal fractures did not appear. The coffee ring showed a larger size, whereas the flakes showed different sizes, being differently arranged near punctual defects. 

In Figure 5A–D, we report the characterization of 2D-MoS_2_ and 2D-WS_2_ NSs exfoliated in cyrene and deposited onto an irregular glass wafer, highlighting the influence of the different substrates, in terms of material and surface properties, on the type of cyrene matrix imaged for the analyzed preparation.

Figure 5A,B present the morphological analyses of MoS_2_ and WS_2_ NSs exfoliated in cyrene film and deposited on an irregular glass substrate support, with the images showing large-scale regions (100 µm in Figure 5A and 20 µm in Figure 5B, respectively) at the border of the deposited drop-glass substrate: the left side shows the drop-casted sample and the right side the glass substrate separated by a red dashed line following the border. It is evident that the film of WS_2_ in cyrene completely filled all the cavities in the glass support, forming a continuous and homogeneous surface (Figure 5B); this only took place partially with the MoS_2_ sample (Figure 5A). In fact, the surface of the MoS_2_ film showed many irregularities that followed the morphology of the glass support, whereas, in the case of WS_2_, only locally were parts of the glass support visible that were not completely covered by the film. The nanoflakes were homogeneously distributed and absorbed in the cyrene layer both for MoS_2_ and WS_2_ and visible only in proximity to deep holes present on the thin film surface (Figure 5C,D). 

It important to stress that, as found from our analyses based on AFM for the thickness of the NSs and on SEM for <L>, as a general feature for LPE in cyrene, the MoS_2_ NSs were bigger than WS_2_ NSs both in thickness and in lateral size. This important finding was ascribed partly to the different pristine bulk sizes of the two materials but could also have been affected by the different intrinsic properties of the crystals. 

In fact, ultra-sonication is a relatively high-energy process resulting in sonication-induced scission during NS exfoliation. This scission effect is actually very important in defining NS dimensions because it plays an important role in reducing the flake size and thickness [56,57]. In our case, we observed larger lateral sizes and thicknesses in 2D-MoS_2_ NSs than in WS_2_ NSs fabricated under comparable exfoliation conditions. The fact that larger flakes tend to be thicker reflects the fact that more energy is required to exfoliate larger-area nanosheets of a given thickness compared to smaller ones [56,58]. This held true in our case because the initial MoS_2_ bulk size (6 µm) was larger than the WS_2_ bulk size (2 µm) as obtained from the producer. 

Additionally, there were differences in the intrinsic properties of the MoS_2_ and WS_2_ bulk crystals that might have affected the final <N> and <L> values of the two materials independently of the solvent choice and exfoliation conditions. Normally, interlayer spacings of 0.615 nm and 0.618 nm exist between the two adjacent layers in MoS_2_ and WS_2_ NSs, respectively, which are connected with weak van der Waals forces [59]. There is also a difference of about 5% between the binding energies of 2D-MoS_2_ and WS_2_ NSs (−0.216 eV and −0.226 eV, respectively), as reported in a recently published study, which is due to the different spatial extensions of the excitons in the intra- and interlayer configurations [60,61]. 

A parameter that was considered as an indicator to evaluate the ease of exfoliation using LPE was the L/N ratio [56,62]. The higher the L/N ration is, the easier it is to obtain high-quality 2D NSs using LPE from pristine bulk materials. For LPE in ethanol, L/N = 11.3 and 10.2 has been measured for the mode values of the parameters for MoS_2_ and WS_2_ and <L>/<N> = 7.1 and 6.9 for the average values for MoS_2_ and WS_2_, respectively [44,46,56], where L is expressed in nm. In cyrene, we can confirm from our experiments that MoS_2_ was easier to successfully exfoliate compared to WS_2_, though the latter certainly showed more indications of the acquisition of quantum dots via LPE. In fact, we estimated that <L>/<N> = 6.5 and 4.3 for MoS_2_ and WS_2_, respectively. We noticed that the <L>/<N> value obtained in cyrene was, however, significantly smaller than that reported in ethanol for the same TMDs.

To conclude this section, the SEM analyses showed the different morphological and textural features of the cyrene film with 2D-MoS_2_ and WS_2_ NSs. The MoS_2_ in cyrene film appeared irregular both in morphological features and in the dispersion of nanoflakes, while the WS_2_ NSs in cyrene film appeared more regular both in morphology and in the dispersion of nanoflakes, as highlighted in Figure 4E,F, showing the experiment with a regular glass support, and in Figure 5A,B, showing the experiment with an irregular glass support.

**Table 1 ijms-24-10450-t001:** Exfoliation of 2D TMDs in organic solvents in comparison with cyrene: main characteristics of the obtained nanoflakes in terms of the principal parameters of the production.

Solvents	Surface Tension (mNm^−1^)	2D Material	Stability (Days)	Lateral Size (nm)	Thickness (nm)	Surface Charge (mV)	References
NMP	40.1	MoS_2_ WS_2_	21 14	340 390–500	1–4.5 2–5	−32.1 −41.1	[63,64]
DMF	37.1	MoS_2_ WS_2_	- 21	220–340 1–2	3–7 ~10	- -	[65,66]
Ethanol–H_2_O	32.98	MoS_2_ WS_2_	~21 30	130–150 ~100	1–3 ~3–4	−22.5 −32	[15,67]
H_2_O	72	MoS_2_ WS_2_	30 7	150–250 700–800	1–3.5 4.56	−27 -	[40,68]
Porcelain	38	MoS_2_ WS_2_	~7 ~7	~2.5 ~1–2	4–5 <5	- -	[69]
Cyrene	33.6	MoS_2_ WS_2_	>30 >60	19–38 11–21	0.9–1.5 0.5–0.9	−50.4 −86.5	Present work

### 2.6. TMD Exfoliation in Cyrene as Compared to Other Solvents: A Comparative Analysis of the Nanoflakes’ Typical Parameters 

In Table 1, we report a comparison of the main parameters related to other commonly used organic solvents with those of cyrene in terms of the exfoliation of the most used 2D TMDs: MoS_2_ and WS_2_.

As we can see in Table 1, the thickness, lateral size and stability of the produced NS dispersions were at quite an acceptable comparative level, making cyrene a potential candidate for the exfoliation of 2D TMDs for various biomedical and electronic applications. Moreover, the stability time of the 2D TMDs exfoliated in the current work in cyrene was longer than with the other reported solvents. This is also a very interesting finding in view of performing various biomedical assays, which require long stability times for a given nanomaterial dispersion. 

Furthermore, the thickness of the 2D TMDs exfoliated in cyrene (in the range of 0.7–1.5 nm) was the lowest, resulting in monolayer and few-layered structures. This is also very interesting, making cyrene the most suitable solvent to obtain the highest possible surface-to-volume ratio, a fundamental parameter in many nanoscience and nanotechnology studies and applications. 

Finally, the highest negative surface charge values for both the TMDs were found in cyrene as compared to the other more spread-out solvents. Such values lead to stronger edge-dangling bond interaction compared to the other solvents reported in Table 1 and could be a very useful feature in view of the NSs’ functionalization with other chemical molecules or assembly with specific chemical groups. 

### 2.7. Surface Charge Analysis: ς-Potential 

Colloidal dispersions are generally stabilized through electrostatic repulsions [70]. In the case of 2DMs, dynamic interactions between the NSs and their electrostatic stabilization play a fundamental role in anticipating the stability of liquid dispersions. MoS_2_ NSs are generally neutral but acquire negative or positive charges upon adsorption of a particular solvent or ionic surfactant [71,72]. Depending on the ionic strength of the solvent or the surfactant, the ς-potential can be measured by assuming the distribution of the ions in the solution. As a consequence, the increase or decrease in the electrostatic potential from the surface of the 2D NSs can be estimated by using well-established equations and physical models, such as the Poisson–Boltzmann equation and the Debye–Hückel model [71,73,74]. In the present scenario, we observed high surface charge densities for MoS_2_ and WS_2_ NSs exfoliated in cyrene. 

The ς-potential values were measured three times for each scan, and the total number of runs was set to 100. Bulk MoS_2_ exhibits high hydrophobicity and, upon exfoliation, the contact angle of the bulk material decreases; after further centrifugation steps, the angle is reduced further and the material will transform from a hydrophobic material to a hydrophilic material in a given solvent [71]. Normally, when exfoliated in pure water, MoS_2_ and WS_2_ NSs exhibit surface charges in the range of −26 mV to −32 mV, which provides a stable dispersion and sufficient electrostatic repulsive forces to avoid the aggregation of dispersed 2D TMDs [38,41,68]. Interestingly, we found that the ς-potential values were significantly more negative in cyrene than those obtained with water for both materials, WS_2_ being significantly more negative than MoS_2_ with values of −86.5 ± 1.5 and −50 ± 3, respectively.

Thus, both MoS_2_ and WS_2_ exhibited high surface charge values. One possible reason for the strong polarity is the formation of hydrophilic groups during the exfoliation process between the 2D NSs and between the polar groups of cyrene and the 2D TMDs. The 2D-MoS_2_ and 2D-WS_2_ NSs were functionalized by ionizable groups (-SH and –HSO_3_) formed during exfoliation and, as a result, the MoS_2_ and WS_2_ NSs became negatively charged by dissociating protons (H^+^). Ultrasonic cavitation not only overcomes interlayer van der Walls interactions to produce thin NSs but is also involved in the process of fragmentation of large flakes into smaller ones. The latter process breaks Mo-S bonds and introduces large amounts of edge-attached dangling bonds, which further react with the solvent molecules. Therefore, cyrene can readily interact with the dangling bonds available on the edges of 2D-MoS_2_ and 2D-WS_2_ NSs, imparting high surface charge density to the dispersion.

On the other hand, the average lateral size of 2D NSs plays an important role in defining the ζ-potential of dispersed NSs. As can be seen from Table 1, the comparison of the different characteristics of cyrene-dispersed 2D NSs with those of other solvents, the average flake size and the viscosity of the solvent affect the ζ-potential of the NS dispersion. The large proportion of dangling bonds on the edges of the 2D-MoS_2_ and WS_2_ NSs dominate over the surface charge of the 2D NSs. The smaller the average flake size, the larger the edge effect is, strengthening the ζ-potential of the dispersed 2D NSs with greater electrophoretic mobility [71,73]. Solvent viscosity and permittivity, in connection with the exfoliation mechanism, impart stability and effective scission to 2D NSs. The 2D-MoS_2_ and WS_2_ NSs added to the viscosity of cyrene and imparted more energy to the suspension, which, via the cavitation effect, delaminated the material. In this respect, cyrene, being a highly viscous solvent, can efficiently transfer the force between the regions of the MoS_2_ and WS_2_ NSs, leading to effective size delamination. Of course, the initial bulk size and centrifuge steps also play equally important roles in defining the final average flake size [75].

### 2.8. Modeling the Interaction between 2D NSs and Cyrene: DLVO Theory

In order to obtain a picture of the interaction between the nanoflakes in cyrene during their fabrication, we used the modified DLVO theory [76] to work out an estimate for the total interaction energy between the nanosheets. We thus calculated the total interaction energy $V^{\mathrm{tot}}$ using the following equation:(2)$$V^{\mathrm{tot}}=V^{\mathrm{EL}}+V^{\mathrm{vw}},$$
where $V^{\mathrm{vw}}$ and $V^{\mathrm{EL}}$ are the van der Waals and the electrostatic interaction energies, respectively. The attractive van der Waals interaction energy was calculated as follows [32]: (3)$$V^{\mathrm{vw}}=-\mathrm{Aa}/12d,$$
where $A={\left(\sqrt{A_{\mathrm{nano}}}-\sqrt{A_{\mathrm{cyrene}}}\right)}^{2}$ is the Hamaker constant [32] relative to the nanosheets machined into cyrene solvent, d is the nanosheets’ reciprocal distance and a is the effective radius of the NS [74]. Based on the values found in the literature, we assumed that $A_{\mathrm{nano}}$ = $29.6\times 10^{-20}$ and $32\times 10^{-20}$ for ${\mathrm{MoS}}_{2}$ and ${\mathrm{WS}}_{2}$, respectively, and $A_{\mathrm{cyrene}}$ = $2.84\times 10^{-20}$ [32,38,77].

The electrostatic repulsive energy $V^{\mathrm{EL}}$, as a function of d, is given by:(4)$$\begin{array}{cc} V^{\mathrm{EL}} & =\pi \epsilon a\left(\zeta^{2}\right)\times \\ & \{\ln[\frac{1+\exp\left(-\mathrm{kd}\right)}{1-\exp\left(-\mathrm{kd}\right)}]+\ln[1-\exp(-2\mathrm{kd})]\}, \end{array},$$
where ζ is the zeta potential of the nanosheets, ϵ is the dielectric permittivity of the solution and k is the inverse of the Debye–Hückel length $\lambda_{B}$, which depends on the temperature and on the ion concentration in the solution and can be calculated as ≈1/0.4 µm for cyrene at room temperature [76]. In our case, we measured the values given in Table 2 for a, ζ.

Based on the above model, we could calculate the total interaction energy between nanosheets in cyrene versus the separate nanosheets, as shown in Figure 6.

Thus, the DLVO theory predicts a similar trend for the two nanomaterials, the reciprocal force of interaction between nanoflakes being repulsive with large separations and attractive for small distances at d < $d_{{\mathrm{WS}}_{2}}$ = 13.5 nm and d < $d_{{\mathrm{MoS}}_{2}}$ = 35.8 nm, respectively, with $d_{{\mathrm{WS}}_{2}}$ and $d_{{\mathrm{MoS}}_{2}}$ being the zeros of the curves, respectively, where the sign of the mutual force reverses. Interestingly, we noticed that these two zero values matched well with the measured <L> for the two materials. The agreement was perfect for WS_2_ (13.5 nm vs. 13 nm) and still quite fair for MoS_2_ (35.8 nm vs. 23 nm). We interpreted this finding as showing that the $d_{{\mathrm{WS}}_{2}}$ and $d_{{\mathrm{MoS}}_{2}}$—the zero values where the NS reciprocal interactions change from attractive into repulsive—could give at least an indication of the lateral size of single nanoflakes obtained in the fabrication, these mutual forces becoming strongly repulsive at d < $d_{{\mathrm{WS}}_{2}}$ = 13.5 nm and d < $d_{{\mathrm{MoS}}_{2}}$ = 35.8 nm, thus setting an onset for the minimum possible size of a single nanoflake.

We also noticed that the barrier energy for transiting from repulsion to attraction was higher and wider for ${\mathrm{WS}}_{2}$ as compared to MoS_2_, which could explain the stronger tendency for MoS_2_ to form aggregates during the fabrication, as observed in the AFM and SEM characterizations.

## 3. Materials and Methods

### 3.1. Exfoliation and Size Selection

The initial, commercialized bulk molybdenum disulphide (MoS_2_) powder (Sigma-Aldrich, Darmstadt, Germany, 69860, particle size: 6 µm, 99%, density: 5.06 g/mL at 25 °C) and tungsten disulfide (WS_2_) powder (Sigma-Aldrich, Darmstadt, Germany, 243639, particle size: 2 µm, 99%, density: 7.5 g/mL at 25 °C) were exfoliated in cyrene as a solvent (Sigma-Aldrich, 807796, molecular weight: 128.13 g/mol, flash point: 108 °C, density: 1.25 g/mL). The materials were dispersed in cyrene at an initial concentration of 5 mg/mL. The solution was pretreated with bath sonication for 15 min. Then, LPE was executed using tip sonication with a Bandelin Ultrasound SONOPLUS HD3200, Berlin, Germany (operating frequency of 20 kHz and maximum power of 200 W) tip sonicator equipped with a probe (KE-76). The pretreated solution was exfoliated for a further 2 h at 60% amplitude using pulse mode (10 s on, 10 s off) in a quartz bottle. To avoid the production of high heat and temperature due to the collision of cavitation bubbles, sonication was performed using an ice bath and the operating temperature of the device was set at 15 °C. Simultaneously, the output power was calculated, and it was consistent throughout the sonication period, resulting in a homogeneous dispersion of the given material. After sonication, the dispersion was centrifuged (Megafuge™ 16 Centrifuge, Thermo Scientific™, Waltham, MA, USA, equipped with a rotating angle rotor) at room temperature at 300 rpm for 60 min and the sediment containing the un-exfoliated part of the material was discarded. The supernatant was moved to 5000 rpm, after which the sediment was discarded in the last step. The supernatant was stored at 4 °C for further use.

### 3.2. Characterization: UV–Vis

To characterize the resulting 2DMs, we used a range of techniques, including ultraviolet–visible (UV–Vis) spectroscopy, Raman spectroscopy, ζ-potential, scanning electron microscopy (SEM) and atomic force microscopy (AFM). These techniques allowed us to assess the thickness, shape and concentration of the resulting materials, as well as their optical properties. The UV–Vis spectroscopy measurements were performed using a Jasco-700 UV–Vis spectrophotometer, Italywith a 1 cm thick quartz cuvette and a spectral range of 200–800 nm to determine the exfoliated MoS_2_/WS_2_ absorption spectra.

### 3.3. Characterization: Raman and AFM

Measurements of Raman spectra were conducted using the commercial WiTec Alpha 300 confocal micro-Raman system. This system is composed of an inverted confocal Raman microscope and an atomic force microscope (AFM) placed on top of the inverted confocal microscope. This latter is equipped with a 532 nm probe generated with a frequency-doubled Nd-YAG laser and a 1800 grove/mm grating, assuring a spectral resolution of ~1.5 cm^−1^. Raman spectroscopy was used to assess the quality of the exfoliated materials by investigating the structural properties of the exfoliated flakes. AFM was used to measure the flake thickness and to investigate the NSs’ surface topography. For this purpose, flake suspensions were first diluted in methanol (1:4 *v*/*v*) and then 50 µL portions of these suspensions were spun on a clean SiO_2_/Si substrate at 3000 rpm for 60 s. Finally, the substrates were washed with acetone and dried with a gentle N_2_ flux. This last step was performed to reduce the embedding of flakes in the cyrene films, which clearly would have prevented the observation of the thinner flakes and the correct evaluation of flake thickness. AFM analysis was performed in intermittent contact mode to avoid the perturbation of flakes’ adhesion to the substrate. For the AFM characterization, nanoflake suspensions were drop-cast on glass coverslips and stored for 48 h at room temperature to allow complete cyrene evaporation. Measurements were performed using a 100× infinite-corrected objective, which provided in-plane and axial resolutions (PSF-HWHM) of ~0.3 and 1 μm, respectively.

### 3.4. Surface Charge and Average Lateral Size Measurement: ζ-Potential

Electrostatic stabilization is an important parameter to analyze the stability of liquid-exfoliated dispersions. The surface charges generated during the exfoliation can be attributed to electrophoretic mobility (μ). The ζ-potential measurements were performed using a Malvern Zetasizer Nano system, United Kingdom (UK) with a He-Ne laser as the excitation source. The ζ-potential measurements were performed to determine the charge on the surface of the exfoliated NMs. All the measurements were carried out at 25 °C.

### 3.5. Morphological Measurements: SEM

A Zeiss Merlin VP Compact SEM apparatus (DiSTAR laboratory, Università degli Studi di Napoli Federico II) was used for morphological analyses of the samples. The Zeiss Merlin VP Compact field-emission electron microscope was equipped with a Gemini II camera and FeSEM: SmartSem software controller. The SEM was composed of three secondary electron detectors (SE2 (classic detector), VPSE (variable pressure) and InlensDuo (low voltage)) and two backscattered electron detectors (AsB and InlensDuo). The SEM was equipped with a charge compensation system and with Oxford Instruments Microanalysis (both EDS X-max 50 and WDS Wave). Data processing was undertaken using INCA version 4.081 (Oxford Instruments (2006): INCA—The microanalysis suite issue 17a C SP1—Version 4.08. Oxford Instr. Anal. Ltd., Oxfordshire, UK); (EDS and WDS) and Aztec (EDS). The Software for data imaging of FeSEM is SmartSem. The samples were placed on a glass support with both smooth and irregular surfaces with an appropriate droplet drop-casting method; then, they were metalized with gold using a sputter coater.

## 4. Conclusions

This study reports the very first experimental characterization of the exfoliation of MoS_2_/WS_2_ nanosheets in cyrene, a dipolar, aprotic, green solvent. liquid-phase exfoliation, which is the most accessible and reproducible fabrication route for large-scale production of nanosheets, was, in our case, employed with some novel steps, such as a pre-treatment phase of bath sonication lasting 15 min, in order to produce smaller-sized and mono/few-layered nanostructures. Consequently, we also observed an enhancement in the efficiency of exfoliation when visualizing the stability of the dispersion for more than a month, which is very meaningful and possibly crucial for various biomedical tests and assays.

UV–visible extinction spectroscopy results showed very clear corroborations with the Raman spectra and Raman mapping of the exfoliated 2D TMD nanosheets when observing the most significant features of the mono/few-layered nanostructures.

SEM morphological analysis revealed peculiar behavior for the nanometric production in cyrene, which was likely due to the high viscosity of the solvent. As a result, a thin, wrinkled, folded and fractured film composed of nanosheets embedded into the solvent was observed with SEM for the 2D-MoS_2_ nanosheets, whereas the 2D-WS_2_ nanosheets showed a very thin film that was less wrinkled and folded and had no fractures, demonstrating a regular morphology. The 2D NSs showed variations in the lateral size according to the film texture. The 2D-MoS_2_ NSs obtained had 122 ± 44 nm of irregular texture and 26 ± 7 nm of regular texture, whereas the 2D-WS_2_ NSs had 43 ± 16 nm of irregular texture and 13 ± 5 nm of regular texture.

The AFM spectra revealed that the thickness of the mono- and few-layered nanostructures were in perfect agreement with the absorbance and Raman spectra.

The zeta-potential characterization of the obtained products revealed that the nanoflakes were negatively charged for both MoS_2_ and WS_2_, with the latter being more negative. For both materials, the absolute value of the measured potential was significantly higher than that typically reported in water exfoliation.

Cyrene, being very viscous and having a high boiling point, posed some serious issues during the exfoliation, post-exfoliation and sample preparation steps for various characterizations, which need to be studied in further detail. Our study, however, demonstrated that the final quality of the nanoflakes in terms of thickness and lateral size was high and comparable (if not even higher, such as for the lowest possible number of layers) with that obtained by using other organic solvents. Considering the supposed green nature of the cyrene solvent, our results indicate it as a potential candidate for numerous biomedical applications involving liquid large-scale production of nanoflakes.

## Figures and Tables

**Figure 1 ijms-24-10450-f001:**
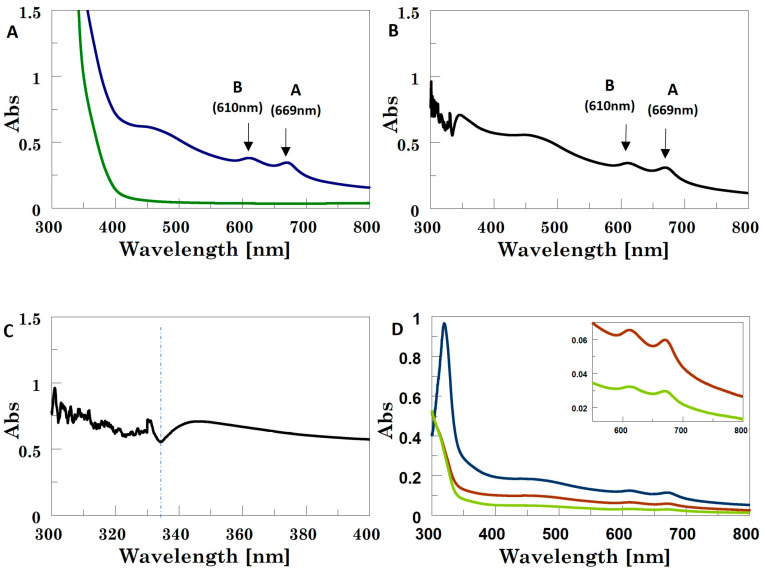
(**A**) UV spectra for cyrene (green line) and MoS_2_/cyrene dispersion (blue line), both diluted with methanol (1:1; *v*:*v*). (**B**) Difference spectrum for 1:1 MoS_2_/cyrene: methanol and 1:1 cyrene: methanol (*v*:*v*). (**C**) Enlargement of the difference spectrum highlighting the position of the minimum at 334 nm. (**D**) Difference spectra for MoS_2_/cyrene: methanol and cyrene: methanol at dilution levels of 1:5 (blue line), 1:9 (red line) and 1:17 (green line). Inset shows the zoomed spectral region from 600 to 800 nm exhibiting the excitonic features of 2D-MoS_2_ NSs.

**Figure 2 ijms-24-10450-f002:**
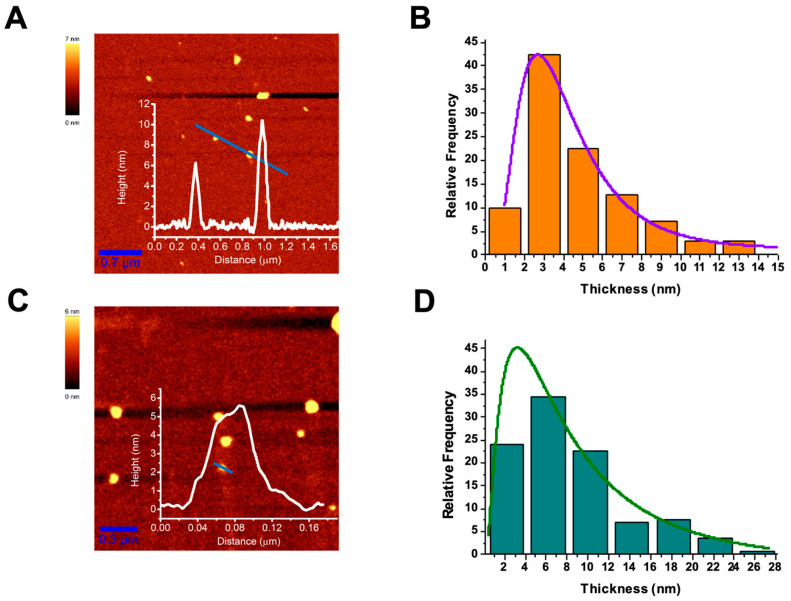
Typical AFM images of WS_2_ (**A**) and MoS_2_ (**C**) nanoflakes acquired in intermittent-contact mode. The two insets correspond to the height profiles across the blue lines shown in the respective topographies. (**B**,**D**) Analysis of flake thickness for WS_2_ and MoS_2_ nanoflakes, respectively. In both cases, the fitting curves of the data with a lognormal distribution are also shown.

**Figure 3 ijms-24-10450-f003:**
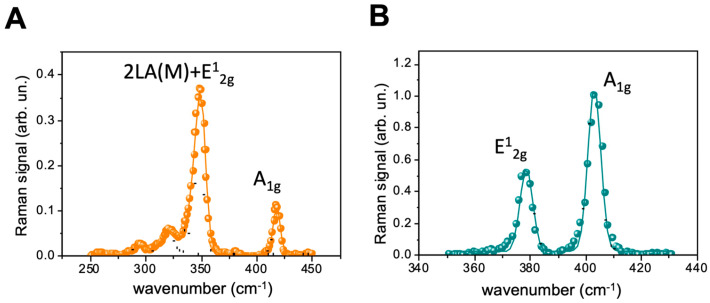
Raman spectra of WS_2_ (**A**) and MoS_2_ (**B**) nanoflakes using 532 nm laser excitation. For both spectra, colored dots correspond to experimental data points, while solid lines correspond to curves fitted with a multipeak Gaussian function. Finally, the single Gaussian peaks are indicated by black dotted lines.

**Figure 4 ijms-24-10450-f004:**
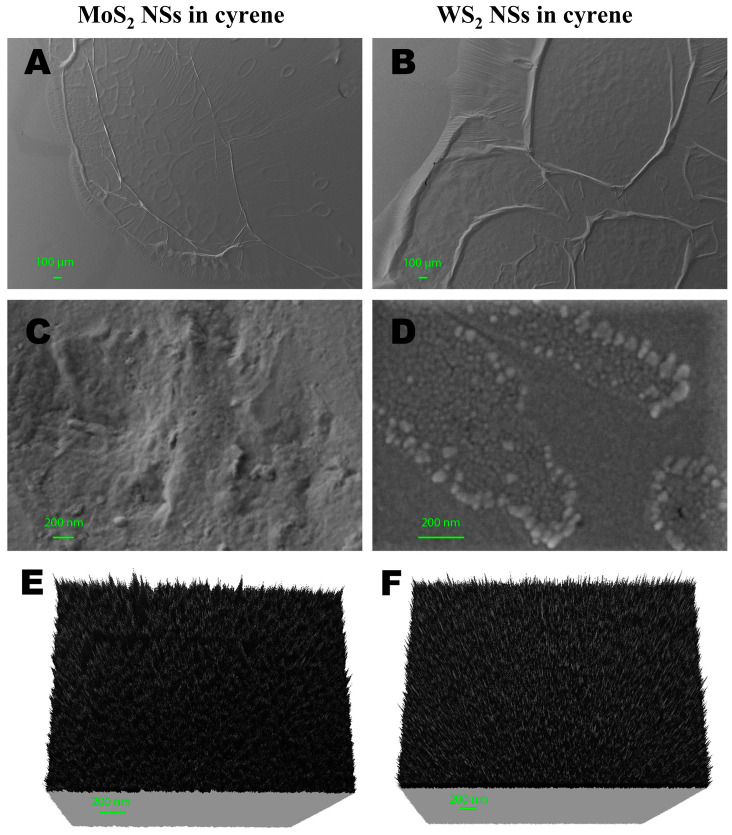
(**A**) MoS_2_ NSs deposited over the silicon substrate absorbed by cyrene at the rim and center of the solid solution. Some coffee ring structures are also visible. (**B**) MoS_2_ NSs chaotically absorbed under the cyrene film at the rim of the solid solution. (**C**) WS_2_ NSs deposited over the silicon substrate absorbed by cyrene on the rim (green arrows) with the largest coffee ring. (**D**) A detailed SEM image of the area with imperfections in the cyrene film. The image shows the presences of WS_2_ NSs arranged in a line and mostly presenting a specific oval-shaped pattern at the periphery. (**E**,**F**) Three-dimensional (3D) maps on a regular surface film (far from the rim) of the MoS_2_ and WS_2_ NSs in cyrene, respectively.

**Figure 5 ijms-24-10450-f005:**
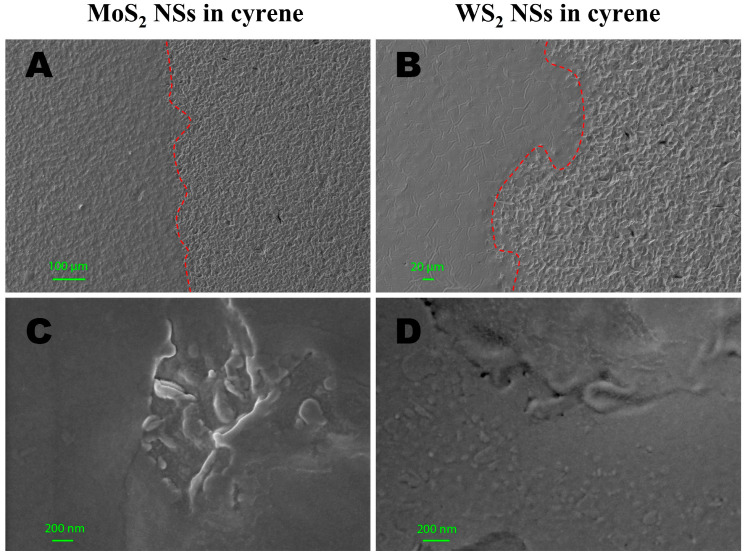
(**A**) Morphological features of MoS_2_ NS film dispersed in cyrene (left) on an irregular glass substrate (right). (**B**) Morphological features of WS_2_ NSs exfoliated in cyrene film (left) on an irregular glass substrate (right). (**C**) Two-dimensional (2D) MoS_2_ NSs deposited and partially absorbed on the irregular surface of the cyrene film. (**D**) Two-dimensional (2D) WS_2_ NSs in the proximity of the irregular surface of the cyrene film.

**Figure 6 ijms-24-10450-f006:**
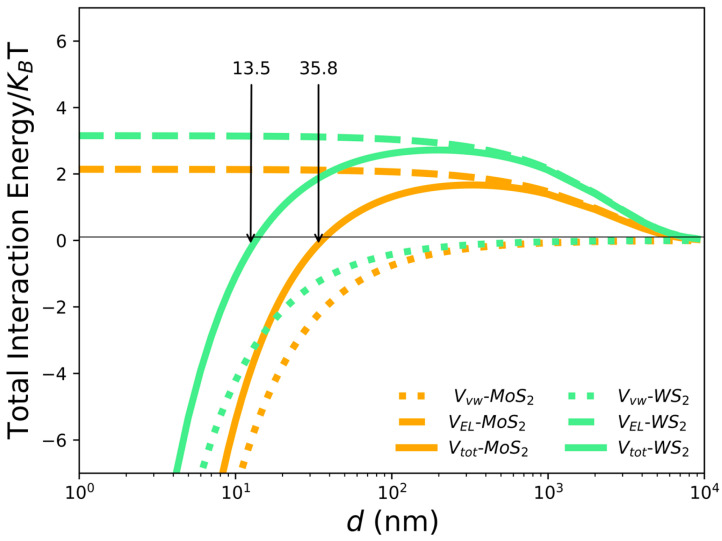
The total interaction energy between nanosheets in cyrene normalized to the thermal energy $K_{B}T$ resulting from the sum of the van der Waals (dotted) and electrostatic (dashed) interaction energies (green for ${\mathrm{WS}}_{2}$ and orange for ${\mathrm{MoS}}_{2}$).

**Table 2 ijms-24-10450-t002:** Measured values of a, ζ for the calculation of the total interaction energy between 2D NSs and cyrene.

Two-Dimensional (2D) TMDs	a (nm)	ζ (mV)
MoS_2_	26	−50.4
WS_2_	13	−86.5

## Data Availability

Not applicable.

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
