# Peer review of "The Effectiveness of Cyrene as a Solvent in Exfoliating 2D TMDs Nanosheets"

_ijms, 2023, doi:10.3390/ijms241310450_

Round 1

Reviewer 1 Report

The paper ijms-2411494 deals with the structural, morphological and optical characterisation od 2D material obtained by liquid phase exfoliation technique by using an environmentally friendly solvent (Cyrene). The topics is surely interesting and the authors present different characterisation like Raman, UV absorption, AFM, SEM and z potential.

Despite the different characterisation technique, the discussion lacks in different point:

line 163 --> how you can give an estimation on the lateral size of the structure from the absorption measurement? Please give more information and add clearly the reference.

In general the discussion is not clear. It could seems that you are just removing the reference to obtain the absorption features of the nanostructure...

Further, it seems that your estimation is 121 nm but in table 1 the measure inserted seems 19-38. Please try to explain better to avoid the confusion for the reader.

Raman spectra There are several papers on the Raman spectrum of 2D materials (MoS2 and MoS2) that can be utilised as references for the experimental finding. The spectral resolution is not indicated (experimental section). the discussion is poor.

The SEM image do not provide any clear information. The resolution is very poor (it should be 10 times higher) and in this form publishable (figure 4 and figure 5).

 liquid phase exfoliation technique 

The quality of English language is adequate

Reviewer 2 Report

In the article The effectiveness of cyrene as a solvent in exfoliating 2D 2 TMDs nanosheets it is presented cyrene as exfoliating agent, to fabricate monolayer and few-layered 2D TMDs. The development and application of environmentally friendly solvents paves the way for reducing hazardous waste and emissions, and it also promotes energy efficiency, safety, and economic viability in chemical process. Thus, the research presented here is of great importance.

Abstract - The abstract is well written but I it requires more specific details about the results. 

Introduction - The introduction is well written and strongly supported by evidence .

Results - There is a part of Figure 2S that says: "absorbance7interference of cyrene..." Is the 7 a typo? 

Table 2 seems displaced.

Overall, the results are well explained and sustained by evidence. 

References - There are few references that does not have the same format. Please verify

The quality of the English Language is good. There are few typos along the text but overall is good. 

Round 2

Reviewer 1 Report

The insets in figure 2a and 2c are not readible. 

Figure 4 and figure 5 are not useful (even with the arrow). Mostly 4b , 4c and 4f + 5 c and 5d. If the authors cannot provide better images (it means with higher magnification), I suggest to remove them from the paper and, if possible increase the discussion.

The quality of the Raman spectra is low  

 Minor editing of English language 

Round 3

Reviewer 1 Report

Ok The authors answered to my indications